# Vertical Transfer of Metabolites Detectable from Newborn’s Dried Blood Spot Samples Using UPLC-MS: A Chemometric Study

**DOI:** 10.3390/metabo12020094

**Published:** 2022-01-20

**Authors:** Alessandra Olarini, Madeleine Ernst, Gözde Gürdeniz, Min Kim, Nicklas Brustad, Klaus Bønnelykke, Arieh Cohen, David Hougaard, Jessica Lasky-Su, Hans Bisgaard, Bo Chawes, Morten Arendt Rasmussen

**Affiliations:** 1Section of Chemometrics and Analytical Technologies, Department of Food Science, University of Copenhagen, Rolighedsvej 26, 1958 Frederiksberg C, Denmark; alessandra.olarini@unimore.it; 2Section for Clinical Mass Spectrometry, Department of Congenital Disorders, Danish Center for Neonatal Screening, Statens Serum Institut, 2300 Copenhagen, Denmark; maet@ssi.dk (M.E.); ACO@ssi.dk (A.C.); DH@ssi.dk (D.H.); 3COPSAC—Copenhagen Prospective Studies on Asthma in Childhood, Herlev and Gentofte Hospital, University of Copenhagen, 2820 Gentofte, Denmark; gozde.gurdeniz@dbac.dk (G.G.); min.kim@dbac.dk (M.K.); nicklas.brustad@dbac.dk (N.B.); kb@copsac.com (K.B.); bisgaard@copsac.com (H.B.); 4Department of Medicine, Harvard Medical School, Boston, MA 02115, USA; rejas@channing.harvard.edu

**Keywords:** transfer, metabolomics, DBS, children, pregnancy

## Abstract

The pregnancy period and first days of a newborn’s life is an important time window to ensure a healthy development of the baby. This is also the time when the mother and her baby are exposed to the same environmental conditions and intake of nutrients, which can be determined by assessing the blood metabolome. For this purpose, dried blood spots (DBS) of newborns are a valuable sampling technique to characterize what happens during this important mother-child time window. We used metabolomics profiles from DBS of newborns (age 2–3 days) and maternal plasma samples at gestation week 24 and postpartum week 1 from n=664 mother-child pairs of the Copenhagen Prospective Studies on Asthma in Childhood 2010 (COPSAC${}_{2010}$) cohort, to study the vertical mother-child transfer of metabolites. Further, we investigated how persistent the metabolites are from the newborn and up to 6 months, 18 months, and 6 years of age. Two hundred seventy two metabolites from UPLC-MS (Ultra Performance Liquid Chromatography-Mass Spectrometry) analysis of DBS and maternal plasma were analyzed using correlation analysis. A total of 11 metabolites exhibited evidence of transfer (R>0.3), including tryptophan betaine, ergothioneine, cotinine, theobromine, paraxanthine, and N6-methyllysine. Of these, 7 were also found to show persistence in their levels in the child from birth to age 6 years. In conclusion, this study documents vertical transfer of environmental and food-derived metabolites from mother to child and tracking of those metabolites through childhood, which may be of importance for the child’s later health and disease.

## 1. Introduction

The time during pregnancy, birth, and the first months of life is crucial for the child’s health and disease. What happens in the woman’s body during the nine months of gestation has long- and short-term effects on physical [1] and psychological [2] health of the child. During pregnancy, the mother and child are exposed to the same nutrients intake and to some extent the same environment. Thus, environmental factors and nutritional status during pregnancy and lactation can have effects on the baby’s health including brain development [3], motor milestones achievement [4], digestive tract development [5], metabolism [6], and immune maturation [7]. Additionally, the life changing event of birth including mode of delivery (vaginal or cesarean section) as well as gestational age, also have lifelong effects [8,9]. For a range of diseases, including asthma, allergy and eczema, predisposition by having a mother with the disease is much stronger than having a father with the same disease [10]. This point towards that the drivers of these diseases constitutes more than genetics [11]. Indeed, pregnancy surrounds the period where the organs resemble, the body establishes, and the immune system develops. Further, in early life the immune system undergoes maturation. This is also the period where the mother is in much more close contact with the child, from carrying the child during pregnancy to breast feeding postpartum, compared to the father. For all these reasons, scientific research is focusing on biochemical and biological processes in women and children during this important time window.

This exploratory study investigates the role of vertical transfer of metabolites from mother to child during pregnancy in a total of 664 mother-child pairs, and if the levels of these transferred metabolites persist to early childhood (6 and 18 month of age) as well as the time around school entry (6 years of age). A longitudinal correlation related mothers at 24 week of pregnancy and their baby ($R_{m24\text{-}c}$), a cross sectional one between mother one week postpartum-child ($R_{m1\text{-}c}$) and a mother-mother correlation ($R_{m24\text{-}m1}$) were computed for investigated metabolites. For the study of vertical transfer, correlation across individuals at the same time point, relating mothers one week postpartum and newborns were used as index. Longitudinal correlation could also be taken into account as a vertical transfer. However, this would imply a metabolite stability of about four months and a time-dependent vertical transfer in comparison to the $R_{m1\text{-}c}$ correlation, which does not have to consider this assumption. Figure 1 shows these correlation analyses and those carried out to study the persistence of transfer up to 6 years of age of the child.

## 2. Results

### 2.1. Identification and Matching of Metabolites

To investigate the vertical transfer, we have matched metabolomics data from DBS and plasma with two different chromatographic systems. The mothers’ dataset consists of 1130 biochemicals, of which 913 were annotated, whereas only 6% of the infants’ metabolome (~150 compounds) was annotated, according to the Metabolomics Standard Initiative’s reporting standards, at level 2 [12]. The identified metabolites were classified as lipids, amino acids, nucleotides, peptides, xenobiotics, cofactors, and vitamins based on the chemical taxonomy provided by Metabolon. Sixty six compounds appeared by the same annotated chemical name between the mothers plasma and newborns DBS datasets. By matching masses in *m*/*z* window of 0.01 resulted in 481 compounds found, including the 66 compounds identified previously. Of the remaining, 98 compounds were found to have a unique match between metabolites detected in mothers and those in DBS, while further 108 were annotated through manual annotation propagation through the Global Natural Products Social Molecular Networking Platform (GNPS) [13], resulting in a total of 272 annotated compounds (Appendix A). These matching compounds are dominated by amino acids (36%), lipids (29%), and xenobiotics (18%). (See Figure 2 for details).

### 2.2. Metabolite Specific Correlation and Transfer

To investigate the correlation between the 272 matched metabolites of mothers and children, Pearson’s correlation was used to estimate correlations within individuals ($R_{m24\text{-}m1}$), across individuals within the same time point (cross-sectional—$R_{m1\text{-}c}$), and across individuals and time (longitudinal—$R_{m24\text{-}c}$). Figure 3 shows the cross sectional mother to child correlation $R_{m1\text{-}c}$ versus the within mother individual correlation $R_{m24\text{-}m1}$. To a large extent, positive correlations are observed. The highest estimated cross-sectional correlations from mother to child ($R_{m1\text{-}c}>0.3$), are shown in Table 1. The cohort undergoes a randomized controlled trial (RCT) intervention with fish oil versus olive oil. In order to investigate the dependency of this intervention on the observed correlation, we have stratified the correlation analysis on each of the two study arms (Appendix A). Particularly, the results for the 3-carboxy-4-methyl-5-propyl-2-furanpropanoate (CMPF) depend on the intervention, and hence the results for the placebo arm are included in Table 1 and visualized in Appendix A. Based on the strength of correlation, we partition the 272 metabolites annotated, to 11 with strong to modest evidence of vertical transfer ($R_{m1\text{-}c}>0.3$), 28 as weakly transferred ($0.3\geq R_{m1\text{-}c}>0.1$) (Appendix A), and 233 non-transferred ($R_{m1\text{-}c}\leq 0.1$) (Appendix A). This partition shows that strong to modest transferred metabolites belong to the class of amino acids, lipids, and xenobiotics, while metabolites considered not to be transferred belong to the class of vitamins, peptides, nucleotides, lipids, xenobiotics, and amino acids. In detail, transferred amino acids are derived from the urea cycle and lysine metabolism, the non-transferred ones from these cycles as well as from those of leucine, isoleucine, valine metabolism and methionine, cysteine and taurine metabolism. Transferred xenobiotics are components of food including coffee and tobacco, while non-transferred xenobiotics derive from the anesthetic and analgesic class, drug metabolites, and also food. Transferred lipids are mainly fatty acids whereas non-transferred ones are long chain polyunsaturated fatty acids (n3 and n6) and fatty acids from monohydroxy and dicarboxylate metabolisms. A comparison of taxonomy transfer-rate is detailed in Table 2. Child metabolite level from DBS as function of maternal blood levels of the 11 metabolites with strong transfer merits are reported to show a more comprehensive view of the correlation trends in Appendix A, as well as correlations between the two maternal time points, which are reported in Appendix A.

### 2.3. Robustness Analysis of the $R_{m1\text{-}c}$ Results

The results for the metabolites exhibiting strong to modest evidence of transfer ($R_{m1\text{-}c}>0.3$), were tested for robustness, by deploying a discovery/replication analysis by the means of bootstrapping. This analysis reveals that all the 11 metabolites are discovered at least 87% of the times, while 8 has a discovery rate of 100%. In the associated independent replication set, these results are replicated at a similar level in at least 72% of the cases, while almost all (99%) replication attempts are nominal significant. (See Appendix A).

### 2.4. Metabolite Transfer or Common Biology?

In order to evaluate if the metabolites exhibiting significant correlations between mother and child can be interpreted as transfer or are merely due to a common underlying process, giving rise to the observed correlations, metabolite to metabolite correlation analysis were performed on the three individual datasets and visualized by PCA, where the loadings were color-coded by the correlation coefficients from $R_{m24\text{-}m1}$, $R_{m1\text{-}c}$ (Appendix A), $R_{m24\text{-}c}$. This analysis shows that the metabolites with high transfer correlations were not exhibiting stronger internal correlations as compared to the correlations between these metabolites and those with no strong signs of transfer (see Appendix A). This is also evident from the PCA as there is no particular grouping of those metabolites in any of the loading plots.

### 2.5. Predicting Child Metabolite Levels from the Totality of Maternal Metabolites

In order to establish whether the totality of maternal metabolites can improve prediction of the single specific metabolites observed in the child beyond just the same metabolite in the mother, multivariate predictive PLS models were built. Each model consists of the maternal metabolomics profile as predictors and the child metabolite level for each specific compound as univariate response (Appendix A). In order to compare the PLS models with the univariate correlations, the cross validated $R^{2}cv$ is used for model evaluation. Overall, these prediction models fail to predict overall metabolite levels in the children, however descent models were found to predict tryptophan betaine ($R^{2}cv$ = 0.68) and CMPF ($R^{2}cv$ = 0.74) from one week postpartum maternal metabolite levels.

### 2.6. Persistence of Transfered Metabolites

For the 272 shared metabolites, the correlation between neonatal DBS levels and childhood levels at 6 months, 18 months, and 6 years of age were calculated. Figure 4 shows these correlation results as a function of the observed correlation between postpartum mother to child ($R_{m\text{-}c}$).

The first analysis investigates homology between vertically transferred metabolites and persistence in the child. This enrichment type analysis reveals correlations of R=0.56, R=0.45 and R=0.46 up to 6 months, 18 months, and 6 years, respectively (all significant at $p<10^{-12}$). This indicates that the metabolites with the strongest vertical transfer results are also the most persistent ones.

Of the 11 metabolites with $R_{m1\text{-}c}>0.3$ in Table 1, 7, 4 and 4 metabolites were exhibiting significant persistence (false discover rate adjusted *p*-value q<0.1) up to age 6 months, 18 months, and 6 years, respectively. These 8 unique metabolites belonged to the biochemical classes of amino acids, xenobiotics, and lipids. Remarkably, the amino acid N6-methyllysine obtained correlations of persistence of R>0.48 within the child, on par with the strength of the vertical transfer, while all other metabolites with indication of transfer had attenuated correlations of persistence.

## 3. Discussion

From a total of 272 matched metabolites obtained in n=664 mother-child pairs, 11 were exhibiting evidence of strong transfer (R>0.3) while a further 28 showed weak—but statistical significant transfer (R>0.1⇒p<0.01). The metabolites with the strongest transfer belonged to the class of amino acids, xenobiotics and lipids. Of the 11 strongest metabolites, 4–7 further showed persistence in the child up to age 6 years.

The metabolome measured using metabolomics techniques is thought to be the most predictive phenotype, reflecting a physiological snap-shot of a biological system at a specific point in time and is able to account for environmental, genetic, and epigenetic factors [14].

Using correlation analysis between three metabolomic profiles measured at gestational week 24, in the child at age 2–3 days and in the mother 1 week postpartum constituted a method for establishing which metabolites show evidence of vertical transfer. Further, the correlations within mother as well as the long term correlations of transfer from mid pregnancy to child postpartum makes up a reference for both the stability in the same body as well as how much can be ascribed to also having shared genetics.

### 3.1. Dietary Metabolites

The furan fatty acid metabolite CMPF exhibited strong evidence of cross sectional transfer. We and others have previously shown that this metabolite is strongly related to the intake of fish [15] and green vegetables [16], and further, that it is transferred to the child [15]. This study adds that indeed these signs of transfer is also evident at the age of 2–3 days. The low correlations observed from mother week 24 to child postpartum is largely due to the fish oil intervention, which is initiated after the week 24 sampling time point.

Tryptophan betaine is a N-methylated form of tryptophan. This metabolite has a higher correlation between individuals, compared to within the same individual over time: $R_{m1\text{-}c}>R_{m24\text{-}m1}$. Since the child has only half the mother’s genetic, a value of $R_{m1\text{-}c}>R_{m24\text{-}m1}$ indicates a strong influence of environment including diet, and tryptophan betaine is also known to be abundant in legumes [17], nuts [18], and in the breastmilk of the mothers after peanut consumption [19].

Our study indicates a high correlation of blood ergothioneine levels between mothers and children, suggesting that it is transferred from mother to child. Ergothioneine can accumulate at high levels in the body from the diet and it is found that its blood levels decline with age. Almost nothing is known about the role of ergothioneine during development in infants, and its role in infant development has begun to be investigated very recently [20].

### 3.2. Metabolites from Smoking and Coffee

Metabolites reflecting coffee intake (caffeine, paraxanthine and theobromine) and smoking (cotinine) were among the vertically transferred metabolites. Cotinine has higher $R_{m24\text{-}c}$ than $R_{m1\text{-}c}$, suggesting that the transfer is highly impacted by the environment and it is not or not only related with the biological and chemical processes, also it is close to theobromine in the loading plot of the PCA model reported in Appendix A, indicating a possible link of the two metabolites in the description of a phenomenon. The transfer of nicotine, cotinine, and caffeine into breast milk in a smoking mother consuming caffeinated drinks has been studied [21]. It can be hypothesized that these two metabolites also reflect smoking and the contextual intake of coffee or tea, which is a common habit. Caffeine has higher $R_{m24\text{-}c}$ than $R_{m1\text{-}c}$ and is highly correlated with paraxanthine, its main metabolite in the loading plot (Appendix A). These metabolites are close in the loading plot of mothers during pregnancy but not in child and mother at postpartum (Appendix A), suggesting a transfer to the baby during pregnancy rather than after delivery.

### 3.3. N6-Methyllysine

N6-methyllysine is a naturally occurring amino acid of the lysine catabolic subpathway found in human biofluids and was recently identified and added to the untargeted metabolomics measurements by Metabolon [22]. For this metabolite we observe a high mother-child correlation of R=0.6, but with an even stronger within mother correlation R=0.82. This could indicate N6-methyllysine to be a metabolite with a strong genetic dependency. Indeed, recently Panyard et al. 2021 [23] shows the eQTL of the PYROXD2 gene to be strongly related to the levels of N6-methyllysine (*p* < 10${}^{-50}$). Further, we observe long-term persistence of this metabolite during childhood, providing supportive evidence for this metabolite being derived from genetic processes.

### 3.4. Non-Transferred Metabolites

There were no correlations of essential nutrients that are immediately used once introduced in the organism, such as metabolites of cofactors and vitamins, nucleotides, amino acids from arginine and proline metabolism, leucine, isoleucine and valine subpathway, methionine, cysteine, taurine, and tryptophan metabolism. The point to note is that the metabolites that are considered transferred belong to the subpathway, which has more non-transferred metabolites, pointing towards that it is indeed the sole molecule that is transferred and not an entire biochemical pathway.

### 3.5. Preprocessing and Matching Metabolomics Datasets

Metabolite annotation was a key challenge for this project. By sampling and storage, dried blood spot blood samples and plasma samples constitute rather different sampling and storage conditions. Furthermore, the analytical procedures for sample preparation and analysis were also following different protocols. Indeed, the number of matched metabolites (272), constitutes only a fraction of metabolites, and hence this analysis likely underestimates the level of vertical transfer. Specifically, oxidation prone metabolites including free fatty acids and phospholipids, as well as certain sugars and amino acids [24,25] may constitute compounds for which limited signal is present in the DBS sample. However, for those metabolites with evidence of transfer, the results are trustworthy or at least only biased downwards.

## 4. Materials and Methods

### 4.1. Study Population

The COPSAC${}_{2010}$ cohort is an unselected mother-child birth cohort including 738 pregnant woman and their 700 children. Recruited woman attended first examinations between pregnancy weeks 22–26 of gestation. Of the 738 pregnant women (average age of mothers at baby’s birth were 32.3±4.3 years), 700 children were enrolled in the study. The children attended the clinic for the first time at age 1 week and then at age 1,3,6,12,18,24,30,36 months, then yearly until the age of 10 [26]. Gestational age was determined on the basis of routine pregnancy care ultrasonography. The study participants of COPSAC${}_{2010}$ included pre-term and post-term delivered infants (30–42 weeks). During the third trimester of pregnancy, the women participated in a factorial-designed, double-blind, randomized controlled trial of high-dose vitamin D (2400 IU/day) or standard dose (400 IU/day) [27] and either 2.4 g n−3 long-chain polyunsaturated fatty acid (LCPUFA, 55% (*w*/*w*) 20:5(n−3), eicosapentaenoic acid (EPA) and 37% (*w*/*w*) 22:6(n−3), docosahexaenoic acid (DHA)) or placebo (72% (*w*/*w*) n−9 oleic acid and 12% (*w*/*w*) n−9 linoleic acid) [28]. Women diagnosed with endocrine, heart, or kidney disease or with a daily vitamin D intake above 600 IU/day were excluded. Children with a gestational age that was less than 32 weeks were excluded.

### 4.2. Ethics

The trial was approved by the National Committee on Health Research Ethics (H-B-2008-093) and the Danish Data Protection Agency (2015-41-3696). Both parents gave oral and written informed consent before enrollment.

### 4.3. Dried Blood Spot Samples Collection and Storage

Dried Blood Spot (DBS) samples for COPSAC${}_{2010}$ newborns were acquired from the Danish National Biobank and described previously in [29]. A few drops of blood (50–75 µL) from the baby’s heel were collected on a pure cotton filter paper card for the newborn screening program for inborn errors of metabolism within the age range of 2–3 days. The blood spots were then stored at −20 °C indefinitely in the Danish Newborn Screening Biobank at Statens Serum Institute (SSI).

### 4.4. Blood Samples Collection and Storage

Blood samples were collected from the mothers at week 24 of pregnancy and 1 week postpartum, and at 6 months, 18 months, and 6 years from the children. The blood sample was collected in an ethylenediaminetetraacetic acid (EDTA) tube (lithium-heparin tube for child 18 months samples) and left at room temperature for 30 min and thereafter spun down for 10 min at 4000 rpm. The supernatant was collected and stored at −80 °C until further analysis [29].

### 4.5. Ultra High Performance Liquid Chromatography—Tandem Mass Spectrometry (UHPLC-MS/MS) Metabolomics Analysis

#### 4.5.1. DBS Sample Preparation

DBS sample (n=682) preparation was conducted at Statens Serum Institute (SSI, Copenhagen, Denmark). The sample preparation was done using the automated liquid handler Microlab STAR^®^ (Hamilton company) using LC-MS (liquid chromatography—mass spectrometry) grade solvents (Thermo Fisher Scientific, Waltham, MA, USA). Metabolites from DBS samples (3.2 mm diameter punch) were extracted onto 96-well plates in 100 µL 80% methanol. The supernatants (75 µL) were transferred onto new plates, dried under nitrogen, reconstituted in 75 µL 2.5% methanol, and transferred (65 µL) onto the final plates before injection. The samples were randomized into 10 batches and were analyzed subsequently. Each plate included 8 water blanks, 1 internal standard mixture, 4 external controls (based on a pooled adult blood sample), 3 paper blanks, 4 pooled samples, 2 diluted pools, and 74 cohort samples. A mixture of 24 isotope-labelled internal standards was added to the extraction solvents for quality control purposes [29].

#### 4.5.2. DBS Metabolomic Profiling

The sample handling and analysis of the dried blood spot samples are described in detail by Gürdeniz et al. 2021 [29]. In brief, the metabolic profiling was conducted at SSI using a Dionex Ultimate 3000 UPLC (ultra performance liquid chromatography) equipped with a Acquity UPLC BEH reverse-phase C18 column (130 Å, 2.1 mm × 50 mm, 1.7 µm) and a Acquity UPLC BEH C18 VanGuard pre-column (130 Å, 2.1 mm × 5 mm, 1.7 µm) (Waters Corporation, Waltham, MA, USA), coupled to a high resolution Q-Exactive Orbitrap mass spectrometer (Thermo Fisher Scientific, Waltham, MA, USA). Tandem mass spectrometric analysis was performed in ESI(+) mode.

The data preprocessing of full-scan spectra was performed using XCMS R package [30] including automated peak detection and alignment algorithms. Preprocessing of full-scan spectra resulted in a feature table in which the relative abundance of each ion in the samples is represented by the peak area. Features are removed if their average abundance among samples are lower than half of the average abundance within the blanks. MZmine version 2.40.152 [31] was used to preprocess MS2 full-scan spectra. MS2 fragmentation spectra and MS2 filtered feature tables were exported. The resulting MS2 filtered feature table were merged with the XCMS preprocessed peak table. The features were assigned to the same feature group if they were eluting within the range of 0.02 min and have high correlation ($R^{2}>0.7$). From each feature group, only the feature with the highest relative intensity was kept. The reduced dataset was used for the subsequent data analysis. To remove inter-batch variation, each feature was divided with the overall mean of its recordings included in the respective batch.

#### 4.5.3. Metabolite Annotation

The aggregated list of MS2 fragmentation spectra was submitted to GNPS and a mass spectral molecular network was created using the feature-based workflow and compared against all GNPS spectral libraries [13,32]. All matches kept between network spectra and library spectra were required to have a mass spectral similarity score of above 0.7 and at least four matched peaks. To further increase the number of annotated metabolites, MS2 profiles of the pooled samples were matched with the mzCloud database using Compound Discoverer version 2.1 (Thermo Fisher Scientific, Waltham, MA, USA).

#### 4.5.4. Blood Metabolomic Profiling of Mothers

Details on sample preparation, UHPLC-MS/MS analysis, and quality control have already been outlined [15]. Untargeted plasma metabolomic analysis on mothers and children blood, was carried out by Metabolon, Inc. (Morrisville, NC, USA) using an ACQUITY UHPLC (Waters, Miliford, CT, USA) QExactive™ Hybrid Quadrupole-Orbitrap™ mass spectrometer interfaced with heated electrospray ionization source (ThermoFisher Scientific, Waltham, Massachusetts, USA) operated at 35,000 mass resolution. Processed samples were analyzed on four platforms: (1) UHPLC-ESI(+)MS/MS optimized for hydrophilic compounds; (2) UHPLC-ESI(+)MS/MS optimized for hydrophobic compounds; (3) reverse phase UHPLC-ESI(−)MS/MS using basic optimized conditions, and (4) HILIC/UHPLC-ESI(−)MS/MS. Metabolites were identified based on three matching criteria: retention time/index range, mass accuracy (±10 ppm), and MS/MS spectra. The compound identification was based on the following criteria: (1) compounds labeled with “*” have identification level 2; (2) compounds labeled with “**” have level 3 (since no standards or matching spectra are available); (3) compounds named with “X-” are unknown and therefore have level 4, and (4) if no label is applied, the identification level is 1 [12].

### 4.6. Datasets

For the primary analysis, three metabolomics datasets were created: women 24 weeks of pregnancy, baby at time of birth, and women 1 week after delivery. A total of 1130 metabolites were included in the women datasets (913 compounds of known identity and 217 compounds of unknown structural identity), and 2908 partially identified metabolic features in the children DBS dataset. Further, 3 datasets from blood samples of the children at 6 months, 18 months, and 6 years of age including 884 compounds of know identity were included for the analysis of metabolomic persistence through childhood (Figure 1). The data can not be shared openly due to person sensitive information, but can be shared upon reasonable request.

### 4.7. DBS-Blood Metabolites Matching

For the investigation of the metabolites vertical transfer, the 2908 partially annotated features in DBS dataset were matched with the 1130 metabolites in the mothers plasma. As the two datasets were acquired using different separation techniques, metabolites were matched using the following criterion:Matched by name: The variable identified in the maternal metabolome dataset and in the dataset of newborns with the same annotated name was considered as a match;*m*/*z* matching: For compounds identified in the maternal metabolome, their exact mass and mass of the adduct ions ${\left[M+H\right]}^{+},{\left[M+Na\right]}^{+},{\left[M-H_{2}O+H\right]}^{+}$ were calculated and compared with the mass of the unknown compounds of the newborns by choosing an *m*/*z* window of 0.01 based on the mass accuracy of the mass spectrometers employed;GNPS confirmation: GNPS metabolite annotations were used for qualitative comparison of tentative compound pairs. The *m*/*z* of a DBS compound was searched in the GNPS database, considering an accepted error of 5–10 ppm. Once the *m*/*z* was identified in the GNPS database, the molecule was inspected in the GNPS network through the visual inspection of MS2 spectral similarity [33].

### 4.8. Data Analysis

#### 4.8.1. Preprocessing and Quality Control

First, missing values (0.03% of data) were assumed to be below lower detection limit and imputed with half of the minimum of each compound. Thereafter normalization (Probabilistic Quotient Normalization, PQN) [34] and autoscaling were applied for the newborns DBS metabolic profiles while the maternal data were only autoscaled.

Quality assessment of analytical acquisition and data processing was performed using ANOVA and principal component analysis (PCA), evaluating variation across the pooled external control samples and for detecting outliers.

#### 4.8.2. Within Maternal and between Mother-Child Correlation Analysis

Correlations of metabolite levels within the mother as well as from the mother to the child were assessed by Pearson’s correlation. A total of three combination were conducted, namely:Correlation within individuals, relating mothers 24 week of pregnancy and mothers 1 week postpartum ($R_{m24\text{-}m1}$);Cross sectional correlation from mother to child within the same week of sampling, relating mothers one week postpartum and children DBS 2–3 days postpartum ($R_{m1\text{-}c}$);Longitudinal correlation across individuals and time, relating mothers 24 weeks of pregnancy and children DBS ($R_{m24\text{-}c}$).

The correlation from mother to child observed postpartum $R_{m1\text{-}c}$, is referred to as the vertical transfer.

#### 4.8.3. Newborn and Childhood Correlation Analysis

The metabolite levels at birth from the DBS samples were correlated with levels at 6 months, 18 months, and 6 years. This is referred to as a persistence analysis. These correlation results are used for an enrichment analysis, where the vector of vertical transfer correlations are compared with the individual vectors of persistence by a correlation analysis. Secondly, the metabolites with the strongest evidence of vertical transfer were evaluated individually for persistence by using significance testing using false discovery rate correction for multiple testing, returning a list of vertically transferred and persistent metabolites at a $q_{value}\leq 0.1$.

#### 4.8.4. Robustness Analysis of Transfer Results

In order to assess the robustness of the observed correlation results between maternal one week postpartum and child DBS, a bootstrap pipeline was developed. By random sampling dyad-pairs with replacement, a bootstrap sample of ∼63% of the dyad pairs as well as an out-of-bag ∼37% sample were constructed to mimic discovery and replication datasets. The bootstrap sample were used to calculate correlations, and to identify metabolites with strong transfer statistics ($R_{m1\text{-}c}>0.3$), while similar correlations were calculated for the replication dataset. The bootstrap procedure is repeated $n_{boot}=1000$ times. From this analysis it is recorded for each metabolite: (1) how many times the bootstrapped correlation is strong, $R_{m1\text{-}c}^{discovery}>0.3$, how many times this correlation replicates at (2) a similar level $R_{m1\text{-}c}^{replication}>0.3$ and (3) nominal significance $p_{m1\text{-}c}^{replication}<0.05$. (See Appendix A).

#### 4.8.5. Multivariate Regression: Partial Least Squares

PLS models were built to predict the concentrations of each metabolite in children individually from the totality of maternal metabolomics. Models were validated using venetian blinds cross-validation with 10 splits. Models were constructed by considering the entire maternal metabolome, and further restricted to metabolites within the biochemical super-pathway from which the response belongs to. The statistical quality of the models was assessed by considering the cross validated $R^{2}$ value for comparison across different metabolites and in reference to the univariate correlation results.

The data were analyzed in MATLAB R2020a (MathWorks, Natick, MA, USA) environment, the algorithms used for the multivariate data analysis were implemented in PLS_Toolbox 8.9 (Eigenvector Research, Wenatchee, WA, USA).

## 5. Conclusions

This study constitutes the so-far largest study of vertical transfer of metabolites from mother to child, characterizing the critical period surrounding pregnancy and birth, in which life-long trajectories of health are established. The work shows the feasibility of deriving the same chemical information from the same biological sample (blood), but with different sampling and storage techniques (plasma and DBS), and with different chromatography procedures. From 272 matched metabolites, 11 showed strong evidence for transfer, while further 28 showed weak transfer. Further, for the majority of these metabolites, the levels were persistent in the child up to 6 years of age. The transferred metabolites belonged to the class of amino acids, xenobiotics, and lipids, and were mostly of environmental and dietary origin. These results constitute preliminary results for further investigation of metabolites association with diseases in young children in future studies.

## Figures and Tables

**Figure 1 metabolites-12-00094-f001:**
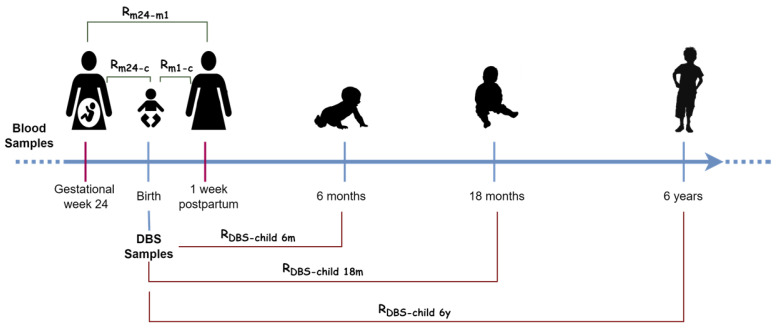
Overview of samples analyzed and correlation analyses performed. Samples examined in the study are: dry blood samples (DBS) for children at birth and blood for women during pregnancy and children at 6 months, 18 months, and 6 years. Correlation analysis for the study of mother-mother temporal stability (n=672) and mother-child vertical transfer from mid pregnancy week 24 (n=664) and 1 week postpartum (n=661) (in green), and correlation analysis for the study of the permanence of transfer from birth to 6 months (n=583), 18 months (n=588), and 6 years of age (n=496) of the child (in red).

**Figure 2 metabolites-12-00094-f002:**
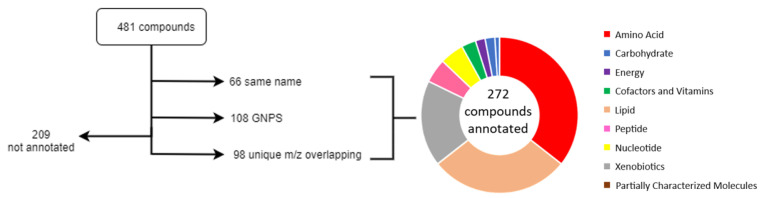
Metabolite annotations based on maternal and child compounds *m*/*z* match and GNPS molecular network investigation. On the right, information on biochemical super pathways.

**Figure 3 metabolites-12-00094-f003:**
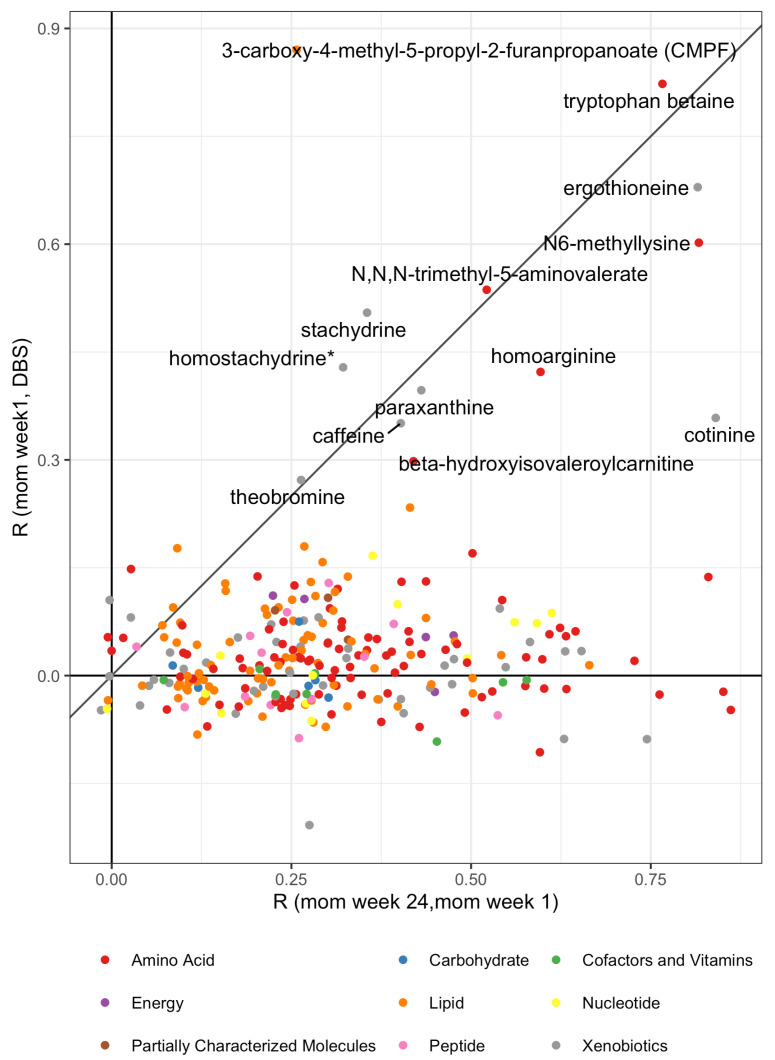
Scatter plot of correlations computed for the 272 metabolites annotated. Correlations of metabolites between mothers ($R_{m24\text{-}m1}$) are reported on the x-axis compared to the correlations between mothers one week postpartum and children at birth ($R_{m1\text{-}c}$) on the y-axis.

**Figure 4 metabolites-12-00094-f004:**
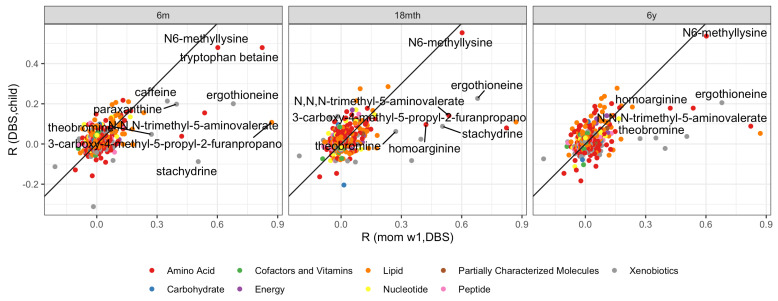
Persistency of metabolite levels from postpartum neonatal (DBS) to child at 6 months, 18 months, and 6 years, as a function of postpartum mother to child $R_{m1\text{-}c}$. Labeled are metabolites with $R_{m1\text{-}c}>0.3$ with a fdr corrected significant correlation (q<0.1). The black line is R(DBS,child)=R(momw1,DBS).

**Table 1 metabolites-12-00094-t001:** Vertically transferred metabolites (R>0.3) with associated biochemical taxonomy and estimated correlations.

Biochemical	Taxonomy	$R_{m24\text{-}m1}$	$R_{m1\text{-}c}$	$R_{m24\text{-}c}$
CMPF **	Lipid	0.26	0.87	0.26
CMPF ** (placebo ${}^{▵}$)	Lipid	0.58	0.60	0.50
tryptophan betaine	Amino Acid	0.77	0.82	0.67
ergothioneine	Xenobiotics	0.82	0.68	0.69
N6-methyllysine	Amino Acid	0.82	0.60	0.56
N,N,N-trimethyl-5-aminovalerate	Amino Acid	0.52	0.54	0.50
stachydrine	Xenobiotics	0.36	0.51	0.41
homostachydrine *	Xenobiotics	0.32	0.43	0.30
homoarginine	Amino Acid	0.60	0.42	0.48
paraxanthine	Xenobiotics	0.43	0.40	0.37
cotinine	Xenobiotics	0.84	0.36	0.48
caffeine	Xenobiotics	0.40	0.35	0.46

*: Putative annotation. **: 3-carboxy-4-methyl-5-propyl-2-furanpropanoic acid. ${}^{▵}$: Analysis from the placebo-arm of the fish oil intervention RCT.

**Table 2 metabolites-12-00094-t002:** Transfer rates by biochemical classes on a total of 272 metabolites analyzed.

Taxonomy	*n*	$n_{R\;>\;0.3}$ (%)	$n_{0.1\;<\;R\;\leq \;0.3}\;$(%)
Amino Acid	98	4 (4%)	10 (10%)
Lipid	78	1 (1%)	11 (14%)
Xenobiotics	48	6 (12%)	2 (4%)
Nucleotide	13	0 (0%)	1 (8%)
Peptide	13	0 (0%)	1 (8%)
Cofactors and Vitamins	8	0 (0%)	0 (0%)
Carbohydrate	6	0 (0%)	0 (0%)
Energy	5	0 (0%)	2 (40%)
Partially Characterized Molecules	3	0 (0%)	1 (33%)

## Data Availability

The data are not publicly available due to participants-level personally identifiable data are protected under the Danish Data Protection Act and European Regulation 2016/679 of the European Parliament and of the Council (GDPR) that prohibit distribution even in pseudo-anonymized form, but can be made available under a data transfer agreement as a collaboration effort.

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
