# Peer review of "Vertical Transfer of Metabolites Detectable from Newborn’s Dried Blood Spot Samples Using UPLC-MS: A Chemometric Study"

_metabolites, 2022, doi:10.3390/metabo12020094_

Round 1

Reviewer 1 Report

Olarini et. al. present a proof a principle study for application of metabolomics in studying non-genetic transfer of metabolites from mother to newborns. Through the comprehensive data, they analyzed the vertical transfer of metabolites from mother to newborn and the persistence of metabolites over the development time course of newborns. The study is interesting with large scale data collection, although no remarkable conclusions were drawn from this study. 

These are some points which I think could improve the readability of the manuscript.

  1. Please introduce the terms at the first usage. For example, in the line 63-64 the correlation coefficients Rm24-m1 etc.  were clear in section 4.8.2. Another example is usage of abbreviations like CMPF, GNPS etc which are not explained at their first usage.
  2. I think section 4.8.2 is important to understand the paper and should come earlier. Vertical transfer definition appears in line 354, which can be explained before.  
  3. Since heavy use of linear statistical analysis is employed throughout the paper, I recommend including scatter plots of 11 metabolites where correlations are computed (m24-m1, m1-c, m24-c). This will also help in demonstrating the variation in concentration level of metabolites between different mother-child pair. 
  4. The caption of Figure 3 can be elaborated. A new term Rm-m appears. Consistency in terminology will be helpful.

I think the authors can also improve the 'Conclusion' section which appears more like a summary of the paper.

Author Response

Response to Reviewer 1 Comments

Olarini et. al. present a proof a principle study for application of metabolomics in studying non-genetic transfer of metabolites from mother to newborns. Through the comprehensive data, they analyzed the vertical transfer of metabolites from mother to newborn and the persistence of metabolites over the development time course of newborns. The study is interesting with large scale data collection, although no remarkable conclusions were drawn from this study. 

These are some points which I think could improve the readability of the manuscript.

1. Please introduce the terms at the first usage. For example, in the line 63-64 the correlation coefficients Rm24-m1 etc.  were clear in section 4.8.2. Another example is usage of abbreviations like CMPF, GNPS etc which are not explained at their first usage.

Response 1.1

Thank you for noting this. We have updated throughout the manuscript accordingly. 

2. I think section 4.8.2 is important to understand the paper and should come earlier. Vertical transfer definition appears in line 354, which can be explained before.  

Response 1.2

We have extended Figure 1 to highlight these analysis, and further expanded the last part of the introduction, to emphasize the methodology. For rigor, we keep 4.8.2 in the data analysis section as well.  

3. Since heavy use of linear statistical analysis is employed throughout the paper, I recommend including scatter plots of 11 metabolites where correlations are computed (m24-m1, m1-c, m24-c). This will also help in demonstrating the variation in concentration level of metabolites between different mother-child pair. 

Response 1.3

Thank you. We have included the scatter plots for the 11 by 3 correlation analysis in the supplementary material (Figure S6 and S7), and further referenced this in the results section. 

4. The caption of Figure 3 can be elaborated. A new term Rm-m appears. Consistency in terminology will be helpful.

Response 1.4

Thank you for capturing this. In line with your comment 1 we have updated the notation. And further extended the figure 3 caption.

“Figure 3: Scatter plot of correlations computed for the 272 metabolites annotated. Correlations of metabolites between mothers (Rm24-m1) are reported on the x-axis compared to the correlations between mothers one week postpartum and children at birth (Rm1-c) on y-axis.” 

5. I think the authors can also improve the 'Conclusion' section which appears more like a summary of the paper. 

Response 1.5

We have rewritten the conclusion to highlight the main findings, and relax some of the not so interesting details. 

Reviewer 2 Report

The manuscript uses metabolomics profiles and correlation analysis to study the vertical mother-child transfer of metabolites. The study has large sample size, long research time and rich research content. The paper has value to be published in Metabolites journal.

However, I still have some questions and suggestions:

  1. I think it would be better if the samples could be divided into discovery and validation sets in metabolomics.
  2. Why is Rm1-c but not Rm24-c referred to as vertical transfer? Thirdly, what does “unique overlap” mean in Line 56?
  3. For the loading plot of PCA, I think there are two points that need to be improved and explained.

(1) The cumulative contribution of the selected principal components is low and does not represent the original data well.

(2) In my opinion, the loadings plot can only see the relationship between variables and variables, and the relationship between variables and samples needs to be seen in conjunction with the score plot.

  1. The author needs to check the manuscript carefully because there are many language errors, including but not limited to the following:

-Line 24: change “is” for” are”. Check the whole manuscript, and there are also many subject-predicate inconsistencies

-Line 32: change “points” for “point”.

-Line 34: change “resembles” for “resemble”. And check Line 34 and 42.

-Section 2.2: Please note the consistency of tenses. Also, what I understand is that R>0.7 means strong correlation but you consider R>0.3 as strong correlation, is that your own definition?

-Line 114 and 121: “r” or “R”?

-Line 118: what does “q” mean?

-Line 251: what does “96 batches” mean?

-Line 254: “96” or “16”?

-Line 255: what does “external controls” refer to?

-Section 4.5.2: missing information on column type.

-Line 350: change “childrens” for “children”.

Author Response

Response to reviewer 2 comments

The manuscript uses metabolomics profiles and correlation analysis to study the vertical mother-child transfer of metabolites. The study has large sample size, long research time and rich research content. The paper has value to be published in Metabolites journal.

However, I still have some questions and suggestions:

1. I think it would be better if the samples could be divided into discovery and validation sets in metabolomics.

Response 2.1

Indeed statistical learning can produce over-optimistic results. However, in this study we do not use parameter-rich modelling paragdigmes, and hence are not torturing the data, making overfitting less of a concern. Further, we want to preserve the study power by using all samples. Merely, the combined setup with more than two timepoints serve as a validation of the results. For this reason we keep the results as is. 

In order to highlight the consistency of the transfer results for the vertical transfer from mother one week post partum to DBS, we have deployed a discovery - validation part to highlight the uncertainty of the results. In this we utilize bootstrapping to split the samples into a bootstrap sample, and the left out ones. 

The bootstrap samples are used to estimate the distribution of the correlation-coefficients. while the out-of-bag samples are used as a replication set. Here we explicitly estimate the correlation, and record whether it replicates the discovery results both at the same level (Rm1-c > 0.3) as well as plain statistical significance (p<0.05 - which is achieved for Rm1-c> 0.125). 

This we have described in the section data analysis “robustness analysis” and  included the results of in the supplementary materials. 

2. Why is Rm1-c but not Rm24-c referred to as vertical transfer? Thirdly, what does “unique overlap” mean in Line 56?

Response 2.2 

For sure, metabolites transferred in mid pregnancy can also be coined vertical transfer, but requires that the metabolites are stable in the child due to a time gap of ~4 month. Hence this time-depend type of vertical transfer comes with additional assumptions as compared to the time-matched  Rm1-c . We have tried to clarify this in section 1, as well we clarify the unique match (overlap) in section 2.1 .

3. For the loading plot of PCA, I think there are two points that need to be improved and explained.

(1) The cumulative contribution of the selected principal components is low and does not represent the original data well.

(2) In my opinion, the loadings plot can only see the relationship between variables and variables, and the relationship between variables and samples needs to be seen in conjunction with the score plot.

Response 2.3

Thank you for raising these points. 

Indeed with a large dataset in both samples and variables, we usually do not observe large variance explained for metabolomics data. We have added an additional analysis in supplementary materials (Figure S5 and Table S6) to underpin the aim of these loading plots: namely to show that the observed transfer results are not due to a single pattern, but that each metabolite can be interpreted individually.

As pointed out sample interpretation is not possible from loading plots only, although we used loading plots to see the relationship between variables.

4. The author needs to check the manuscript carefully because there are many language errors, including but not limited to the following:

-Line 24: change “is” for” are”. Check the whole manuscript, and there are also many subject-predicate inconsistencies

-Line 32: change “points” for “point”.

-Line 34: change “resembles” for “resemble”. And check Line 34 and 42.

-Section 2.2: Please note the consistency of tenses. Also, what I understand is that R>0.7 means strong correlation but you consider R>0.3 as strong correlation, is that your own definition?

-Line 114 and 121: “r” or “R”?

-Line 118: what does “q” mean?

-Line 251: what does “96 batches” mean?

-Line 254: “96” or “16”?

-Line 255: what does “external controls” refer to?

-Section 4.5.2: missing information on column type.

-Line 350: change “childrens” for “children”.

Response 2.4 

Yes,  we consider 0.3 a good correlation in metabolomics taking into account that we consider different people at different times. Thank you for your detailed review. We have updated the manuscript accordingly. 

  • Concerning the numbers, the paper by Gurdeniz et al describes both cohorts, while we in the present study only use COPSAC2010, where 10 batches were used. 
  • The external control samples are based on a pooled adult blood sample repeated across all plates.
